# Alkali metal bilayer intercalation in graphene

Yung-Chang Lin [1,2] ✉, Rika Matsumoto[3], Qiunan Liu[2], Pablo Solís-Fernández [4], Ming-Deng Siao[5], Po-Wen Chiu [5,6], Hiroki Ago [4,7] & Kazu Suenaga [1,2] ✉

Alkali metal (AM) intercalation between graphene layers holds promise for electronic manipulation and energy storage, yet the underlying mechanism remains challenging to fully comprehend despite extensive research. In this study, we employ low-voltage scanning transmission electron microscopy (LV-STEM) to visualize the atomic structure of intercalated AMs (potassium, rubidium, and cesium) in bilayer graphene (BLG). Our findings reveal that the intercalated AMs adopt bilayer structures with *hcp* stacking, and specifically a $C_6M_2C_6$ composition. These structures closely resemble the bilayer form of *fcc* (111) structure observed in AMs under high-pressure conditions. A negative charge transferred from bilayer AMs to graphene layers of approximately $1$~$1.5 \times 10^{14}$ e$^-$/cm$^{-2}$ was determined by electron energy loss spectroscopy (EELS), Raman, and electrical transport. The bilayer AM is stable in BLG and graphite superficial layers but absent in the graphite interior, primarily dominated by single-layer AM intercalation. This hints at enhancing AM intercalation capacity by thinning the graphite material.

Intercalation, the insertion of foreign atoms or molecules within a host material, has garnered significant attention for its potential applications in energy storage, catalysis, and electronics[1–3]. The weak interlayer interaction in layered materials creates a natural two-dimensional (2D) nano-container capable of hosting foreign species for self-assembly[4]. The intercalation of AMs in graphite has been extensively studied for nearly a century since its initial report in 1926 by K. Fredenhagen and G. Cadenbach[5]. Alkali metals (AMs), including lithium (Li), sodium (Na), potassium (K), rubidium (Rb), and cesium (Cs), have attracted particular interest due to their low electronegativity, high reactivity, catalytic behavior, and ability to store charge in graphite[4]. The AM-graphite intercalation compounds (AM-GICs) exhibit tunable optical properties, transitioning from black to gold in color as the intercalation density increases[6]. The transfer of charge from AMs enhances the carrier concentration in the graphene layers, resulting in GICs with high conductivity and even superconductivity[7–10]. The application of AM-ion batteries is particularly significant in this era of

high energy consumption in which the pursuit of increased charge capacity remains crucial.

To gain a deeper understanding of GIC devices and enhance their properties, it is essential to elucidate the arrangement of AM atoms between graphene layers. However, the atomic structure of AMs intercalated in graphite remains unclear, primarily due to the challenge of directly visualizing the intercalants. Previous studies have relied on X-ray diffraction (XRD)[4] and electron diffraction techniques[11] to infer the intercalation stage and atomic arrangement. Based on XRD measurements, it has been widely accepted that each individual interlayer space within the GIC can accommodate only a monolayer of AM atoms. However, these methods have limited spatial resolution, hindering atomic-level characterization. Recently, a study utilizing transmission electron microscopy (TEM)[12] and density functional theory (DFT) calculation[13] indicated the presence of multilayer stacking of Li within the interlayer gap of BLG. These findings suggest that the number of

[1]Nanomaterials Research Institute, National Institute of Advanced Industrial Science and Technology (AIST), Tsukuba 305-8565, Japan. [2]The Institute of Scientific and Industrial Research (ISIR-SANKEN), Osaka University, Osaka 567-0047, Japan. [3]Department of Engineering, Tokyo Polytechnic University, 5-45-1 Iiyamaminami, Atsugi, Kanagawa 243-0297, Japan. [4]Global Innovation Center (GIC), Kyushu University, Fukuoka 816-8580, Japan. [5]Department of Electrical Engineering, National Tsing Hua University, Hsinchu 30013, Taiwan. [6]Institute of Atomic and Molecular Sciences, Academia Sinica, Taipei 10617, Taiwan. [7]Interdisciplinary Graduate School of Engineering Sciences, Kyushu University, Fukuoka 816-8580, Japan. ✉e-mail: yc-lin@aist.go.jp; suenaga-kazu@saken.osaka-u.ac.jp

graphene layers may play a significant role in determining the capacity to store AM atoms.

The formation and dynamic behavior of the intercalants within the confined interlayer space can be significantly different from their 3D counterparts. However, without the ability to visualize their atomic-level structures, these distinctions would remain elusive. BLG is a promising host substrate for visualizing intercalant structure using LV-(S)TEM[14], as it lacks the additional blocking layers found in thicker graphite. In this study, we aimed to unravel the intercalation behavior and the storage capacity of BLG and graphite for AM intercalation. In contrast to the electrochemical intercalation process, vapor phase intercalation represents a relatively "clean" process and can allow a clear visualization of the atomic structure of AM in BLG. Our focus was primarily on K, Rb, and Cs due to their larger atomic size, which makes them easier to detect and image using STEM. Our observations reveal that the intercalated AMs adopt a bilayer configuration with a *hcp* structure within the interlayer gap of BLG. No single-layer or three-layers AM configurations were found in BLG. The interlayer spacing can be expanded to up to 11.25 Å by the presence of two layers of Cs atoms. This bilayer AM structure is also observed in thin graphite films with a thickness of 5 nm but not in thicker graphite films. The amount of charge transfer from the intercalated AMs to BLG is further measured by EELS, providing valuable insights into the electronic interaction between the intercalant and the host material.

## Results and discussion
### Cs intercalation in bilayer graphene

The intercalation of AMs (K, Rb, and Cs) into the BLG on TEM grids was achieved through a vapor phase intercalation process. We began our analysis with the intercalation of Cs, the heaviest AM. The number of synthesized graphene layers[15,16] was determined by the reflecting color contrast after transferring to a $SiO_2$/Si substrate, along with corresponding Raman spectroscopy. The quality of the BLG sample was extensively studied, and growth parameters were fine-tuned to achieve nearly 95% BLG coverage with less than 1% of trilayer or multilayer graphene in the grown samples. To evaluate if the sample is full intercalated, a large pyrolytic graphite sheet (PGS) measuring 0.5 cm × 2 cm × 25 μm was placed in a glass ampoule alongside the BLG sample on a TEM grid (0.3 cm in diameter). The PGS serves as an indicator for monitoring the intercalation stage. When the PGS turns a golden color, it signifies that the intercalation has reached stage 1 for the graphite, marking the endpoint of the reaction. Given that the BLG sample shares the same environment as the PGS, and its relatively smaller size compared to the PGS, BLG should also meet the conditions for stage 1 full intercalation (details are presented in the Method section). Figure 1a presents an annular dark-field (ADF) image of Cs intercalated in BLG. On the left side, BLG is observed with isolated Cs atoms anchored, and the twist angle between the two graphene layers is determined to be 2.7° by analyzing the fast Fourier transform (FFT) pattern (Fig. 1b). The right half of the ADF image displays a large and continuous domain of Cs, showing a honeycomb lattice with hexagonal symmetry, as confirmed by the FFT image shown in Fig. 1c. The number of graphene layers can also be precisely determined based on the ADF contrast, and the intercalated Cs is confined within the BLG region, as shown in Supplementary Fig. 1. The first-order diffraction spots correspond to the Cs[100] plane, indicating a d-spacing of 3.70 Å. The second-order diffraction spots, corresponding to the Cs[110] plane, exhibit a d-spacing of 2.14 Å, which is commensurate with the graphene C[100] spots. Based on the structural configuration, it can be concluded that the Cs atoms form a $\left(\sqrt{3}\times\sqrt{3}\right)R30^{\circ}$ lattice. At first sight it can be consistent to the $C_6CsC_6$ structure overlaid on the graphene lattice illustrated in Supplementary Fig. 2b. However, the Cs lattice in single-layer $C_6CsC_6$ structure does not exhibit the honeycomb lattice observed in the experimental real space image. Producing a honeycomb Cs lattice requires having two Cs atoms in the

unit cell, forming a $C_6Cs_2C_6$ structure, as depicted in Fig. 1b and Supplementary Fig. 2c. The two Cs lattices with identical $\left(\sqrt{3}\times\sqrt{3}\right)R30^{\circ}$ structure are represented by different shades of pink for better visualization. The chemical composition of the $C_6Cs_2C_6$ is further confirmed by EELS quantification, as presented in Supplementary Fig. 3. This raises the question of whether the Cs atom in this super-dense $C_6Cs_2C_6$ structure arrange itself in single-layer form, or should separate into two layers with two individual atomic planes.

To address this question, we employed both experimental and theoretical approaches. In the ADF image shown in Fig. 1a, a clear distinction can be observed between the upper and lower domains in terms of atomic arrangement, as well as in the corresponding FFT pattern (Supplementary Fig. 4). The pink-colored spheres overlaid on the Cs atoms of the two domains indicate a hexagonal arrangement for the upper domain, while the lower domain exhibits a laterally displaced atomic configuration along the x-axis. Figure 1f illustrates the ADF profile of these two domains, obtained from the pink and light green boxes in Fig. 1a. The shortest distance between neighboring Cs atoms in the honeycomb Cs lattice of the upper domain measures approximately 0.25 nm, which is consistent with the atomic model of $C_6Cs_2C_6$ structure shown in Fig. 1d. In contrast, the lower domain exhibits a shorter Cs-Cs distance of 0.14 nm. The nearly 56% reduction in the Cs-Cs distance along one direction is highly unusual for a monolayer system under equilibrium conditions at room temperature, without any external strain. Our experiments therefore revealed multiple Cs intercalated domains in BLG displaying distorted structures with inequivalent shortest Cs-Cs distances (Supplementary Fig. 5). The migration of Cs domains with the same structural configuration was also monitored during in-situ STEM imaging. Our analysis indicates that this migration can be attributed to the displacement of two individual Cs lattices (Supplementary Fig. 6). This indicates that the two Cs lattices separate into two layers along the c-axis, as illustrated in the model shown in Fig. 1e. Our DFT geometry optimization calculations also support this bilayer structure model, as the total energy of K, Rb, and Cs intercalated in BLG shows lower energy for the bilayer form compared to the monolayer (Supplementary Fig. 7). The interlayer spacing in bilayer graphene is 3.4 Å, which expands to 5.95 Å with the intercalation of one Cs layer and to 11.25 Å with two Cs layers. The two layers of Cs exhibit a close-packed stacking structure, allowing for a slight lateral movement and resulting in distorted hexagonal patterns where Cs atoms appear dimerized. This *hcp*-stacked Cs bilayer in BLG resembles the bulk Cs crystal under high pressure conditions, which adopts an *fcc* structure when viewed from the (111) direction (Supplementary Fig. 8). The distance between two Cs layers in the bulk Cs crystal along the *fcc*(111) plane is approximately 3.45 Å. After intercalation into BLG, Cs metal atoms tend to undergo electron charge transfer to the outer graphene layers, resulting in a partial positive charge on each Cs atomic layer. Consequently, this leads to an enlargement of the distance between the two Cs layers to 5.02 Å due to the repulsive electrostatic forces involved, therefore resulting in further expansion of the BLG interlayer distance. At room temperature and ambient pressure condition, AMs are known to crystallize in a *bcc* structure, while they tend to transform into *fcc* structure under high pressure[17–20]. For Cs, the *bcc*-to-*fcc* phase transformation occurs at pressures ranging from 2.37 to 4.3 GPa[21–23]. Therefore, the shortest Cs-Cs hcp Cs bilayer distance in projection is calculated to be 2.44 Å which is nearly identical to that of Cs bilayer found in this study. This finding confirms that spatial confinement between BLG layers provides a two-dimensional (2D) space for Cs atoms to stack and form high-pressure phase with a bilayer configuration. Previous predictions have also indicated that molecules trapped in the van der Waals gap between two graphene layers can experience pressures as high as 1 GPa[24]. Interestingly, we also found that the domain size of the intercalated Cs bilayer varies with the twist angle of BLG. This will be discussed in detail in a follow-up paper.

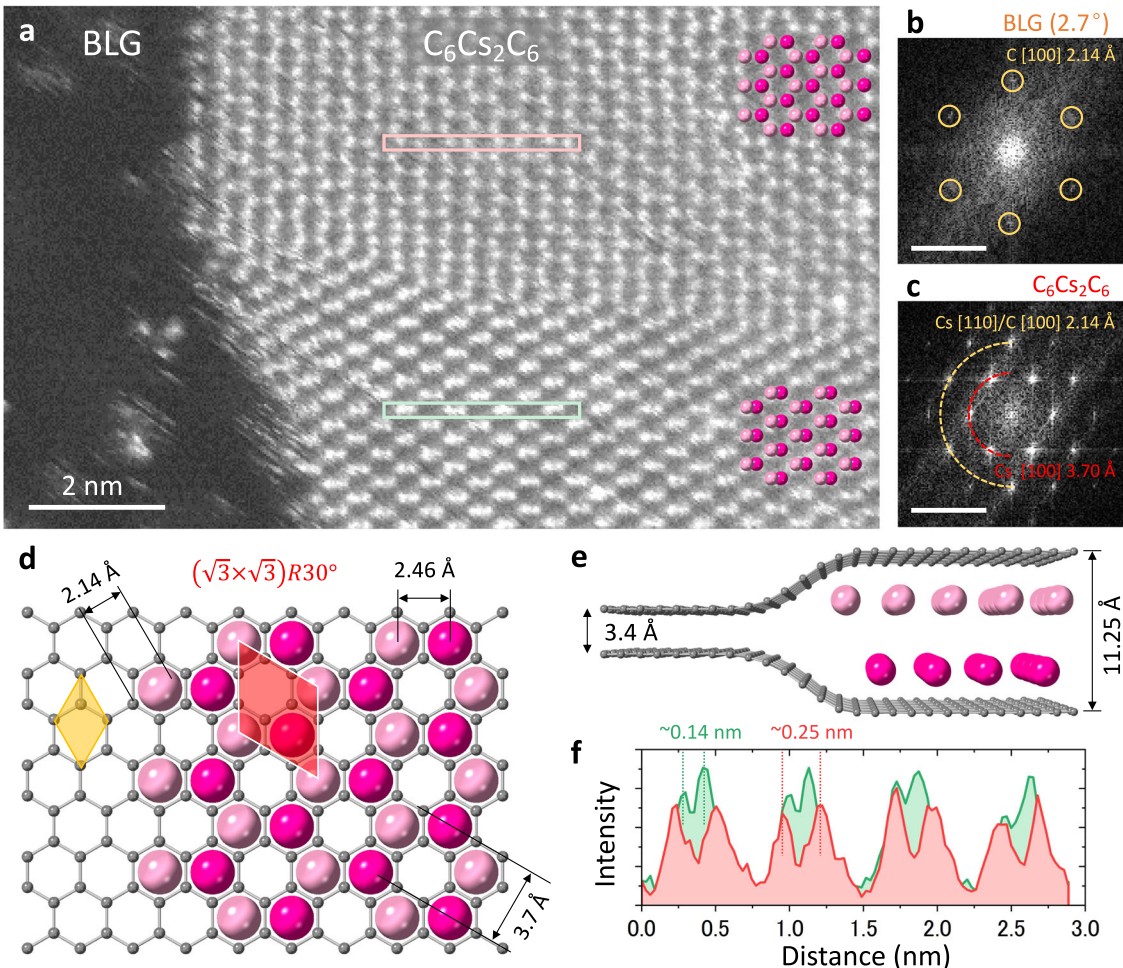

**Fig. 1 | Scanning transmission electron microscopy (STEM) image and atomic structure of Cs bilayer in bilayer graphene (BLG) ($C_6Cs_2C_6$). a** A STEM annular dark-field (ADF) image of Cs-intercalated BLG displaying the $C_6Cs_2C_6$ structure. **b**, **c** Fast Fourier transform (FFT) image of BLG and Cs domains in (**a**) (scale bar = 5 nm$^{-1}$). BLG exhibits a twist angle of 2.7° between its two graphene layers. The [100] and [110] spots of $C_6Cs_2C_6$ domain correspond to interplanar d-spacings of 3.70 Å and 2.14 Å in real space, with the Cs [110] spots overlapping with the graphene [100] spots. **d** Top-view atomic model of $C_6Cs_2C_6$. The yellow rhombus highlights a graphene (1 × 1) unit cell with a = b = 2.46 Å, while the red rhombus highlights the unit cell of the Cs lattice with $\left(\sqrt{3}\times\sqrt{3}\right)R30°$ lattice, where a = b = 4.26 Å. **e** Side and perspective view of the $C_6Cs_2C_6$. The Cs atoms in different atomic planes are color-coded with two different shades of red. **f** ADF profile of the hexagonal Cs layer along the pink and light green boxes in (**a**). The red profile displays the shortest distance between two Cs atoms as 0.25 nm, consistent with the atomic model shown in (**d**). The green profile shows the shorter distance between the Cs atoms, indicating the lateral displacement of the two Cs atomic plans.

## Charge transfer of alkali metal intercalated BLG

Similar to Cs, the intercalation of K and Rb in BLG was also performed and examined using STEM. Compared to Cs intercalation, K and Rb were found to be more sensitive to the electron beam. Lowering the probe current to 5 pA was a key step to capture the ADF image of K and Rb in BLG. Our observations revealed that both K and Rb exhibit the same *hcp* bilayer arrangement as Cs, forming a $C_6M_2C_6$ structure, as shown in Figs. 2a, c. Single crystal BLG[15] and polycrystalline BLG[25] were both employed for AM intercalation, and we observed a $C_6M_2C_6$ structure in both cases. Consequently, both edge and defect inter-calation processes should presumably contribute to the overall inter-calation of AM in BLG. Essentially, we determine whether the AM bilayer resides in the interlayer gap or adheres to the graphene surface based on structural stability under e-beam irradiation and spectro-scopy. The AMs adsorbed on the surface are mostly oxidized and amorphous; even if they initially form a crystalline structure, the sur-face AM layers are simply wiped out by the e-beam. However, AMs intercalated in BLG show lattice defects such as multiple vacancies during prolonged scanning e-beam observation, as demonstrated in Supplementary Fig. 9. The intercalated AM bilayers are consistently

confined within the region of the BLG domain after the e-beam damage to the sample (see also Supplementary Fig. 1), suggesting that the AM bilayer is confined in the interlayer gap between two graphene layers. More interestingly, the intercalated AM layer is uniformly distributed in BLG (Supplementary Fig. 10), and we have not observed single-layer, three-layer, or four-layer arrangements of these AM layer in BLG. We conducted STEM image simulations for 1–4 layers of AM, as illustrated in Supplementary Fig. 11. Through the quantitative ADF contrast comparison, we can effectively distinguish between 2 layers from 3 and 4 layers. The results of EELS quantification for K and Rb, confirming the $C_6M_2C_6$ structure, are summarized in Supplementary Fig. 12. Based on our findings, we can confidently conclude that the intercalated AM consistently adopts a bilayer form. This consistent bilayer form across all three heavy AMs suggests a common structural pattern for inter-calation, with the $C_6M_2C_6$ structure being the most stable configura-tion in BLG. These bilayer K, Rb, and Cs in BLG are well protected from oxidation by the graphene layers. The low electronegativity nature of AM tends to donate electron when in contact with other elements, contributing to stabilization. In the intercalation of two AM layers, both layers can undergo charge transfer to the outer graphene layers.

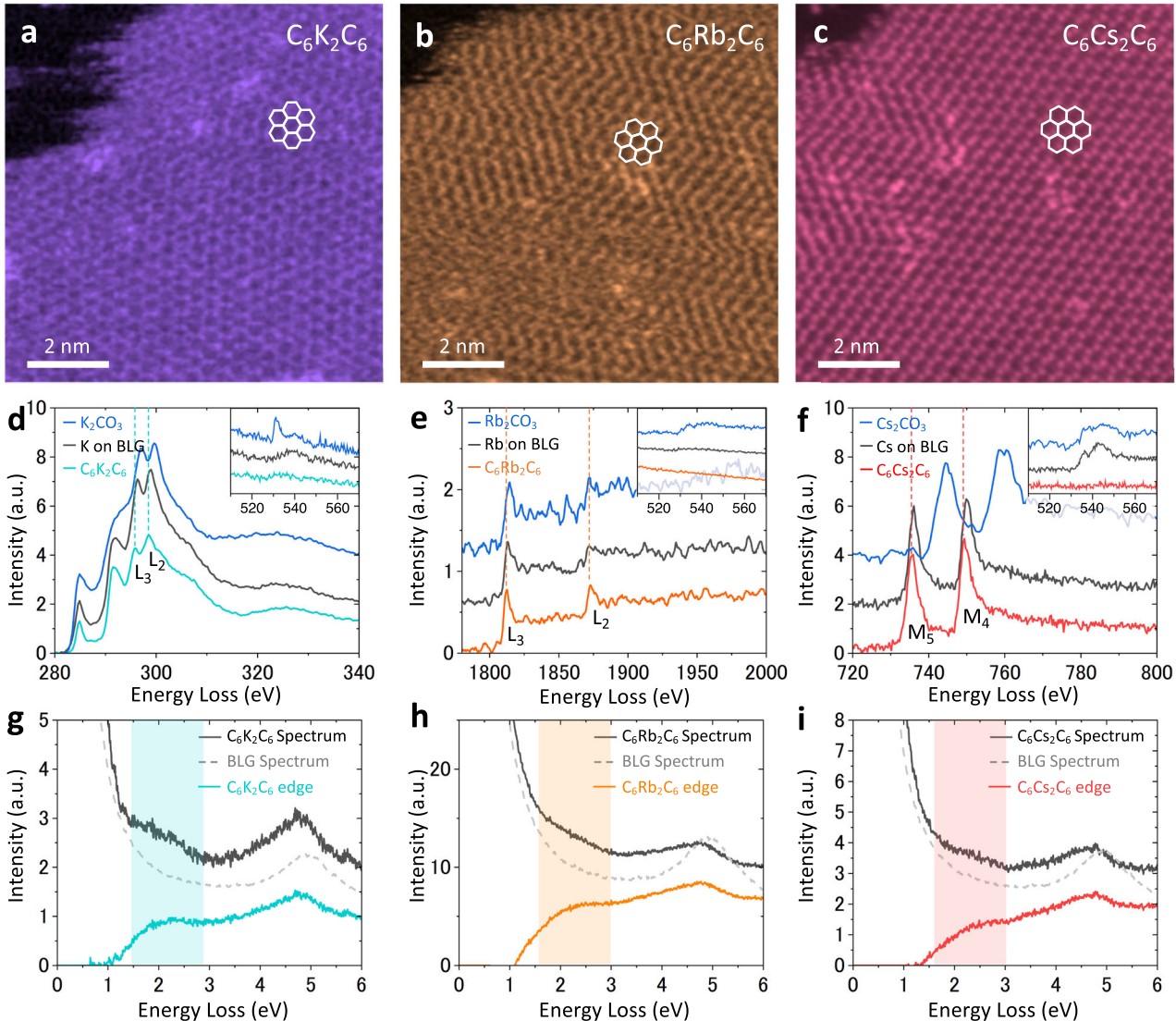

**Fig. 2 | Structure and spectroscopy of K, Rb, and Cs bilayers in BLG. a–c** ADF image of $C_6K_2C_6$, $C_6Rb_2C_6$, and $C_6Cs_2C_6$ in BLG. **d** The electron energy loss spectroscopy (EELS) core loss spectra for the K $L_{2,3}$-edge of $C_6K_2C_6$, amorphous oxidized compounds of K adsorbed on the surface of BLG, and $K_2CO_3$. **e** The EELS core loss spectra for the Rb $L_{2,3}$-edge of $C_6Rb_2C_6$, amorphous oxidized compounds of Rb adsorbed on the surface of BLG, and $Rb_2CO_3$. **f** The EELS core loss spectra for the Cs $M_{4,5}$-edge of $C_6Cs_2C_6$, amorphous oxidized compounds of Cs adsorbed on the surface of BLG, and $Cs_2CO_3$. The oxygen K-edges for these materials are presented in the insets. **g–i**, EELS valence loss spectra of K, Rb, and Cs intercalated samples. The black lines represent the original valence loss spectra, while the colored lines depict the background-subtracted edges. The grey dashed lines serve as a reference spectrum of AB stacking BLG. The charge carrier plasmon in all three samples appears as a broad peak around 1.5 ~ 3 eV, as highlighted by the corresponding colors.

However, if trilayers or more layers of AM are intercalated, no charge transfer will occur in the interior layers of AM, result in less energetic favorability compared to the bilayer case. The theoretical calculation[13] also suggests that the trilayer form of K, Rb, and Cs is less energetically favorable than the bilayer structure when intercalated in BLG. Additionally, an increasing number of AM layers also results in less favorability in formation energy in BLG.

Surface absorption is an inherent aspect of the gas phase intercalation sample. However, even when the BLG is coated with amorphous or oxidized AMs, the intercalated structure maintains the consistent bilayer form of the $C_6M_2C_6$ structure. In Supplementary Fig. 13b, an ADF image of Rb-intercalated BLG with surface deposition is presented, revealing the crystalline structure of Rb in BLG. However, due to the presence of a surface amorphous layer, the imaging quality is somewhat diminished (blurred FFT spots of Rb in BLG). To address this, we subjected the sample to an e-beam shower for 5 min over a large area (~ 100 μm²). This treatment effectively removed surface-

deposited AMs or their oxides (Supplementary Fig. 13a), and the crystalline intercalated AMs remained and were clearly visualized in Supplementary Fig. 13c. Based on these observations, we conclude that surface deposition does not adversely affect the intercalated structure. We used EELS to investigate the oxidation behavior of AM bilayers in BLG, comparing them with the oxidized AMs on the surface of BLG, as well as reference oxides ($M_2CO_3$). The results are displayed in Figs. 2d–f. The $L_3$ and $L_2$ edges of K in BLG ($C_6K_2C_6$) show peaks at 295.8 and 298.5 eV, respectively. The oxidized clusters exhibit an amorphous structure with covalent bonds to hydrocarbons, leading to a shift in the K-$L_{3,2}$ edges to 296.3 and 298.9 eV. In the case of $K_2CO_3$, the K-$L_{3,2}$ peaks shift to higher energies at 297.1 and 298.5 eV. For Rb, the Rb-$L_{3,2}$ peaks are less sensitive to the oxidation state compared to K. The Rb-$L_3$ peak for $C_6Rb_2C_6$ is observed at 1812.4 eV, exhibiting a blue shift of 1 eV for oxidized Rb absorbed on BLG surface, and another 1 eV shift in the case of $Rb_2CO_3$. As for Cs, the Cs-$M_{5,4}$ edges of $C_6Cs_2C_6$ are observed at 735.6 and 749.5 eV. The Cs-$M_{5,4}$ peaks for oxidized Cs

on BLG exhibit a shift to 736.1 and 750.1 eV and to 744.6 and 759.5 eV in the case of $Cs_2CO_3$. These results indicate that BLG can effectively protect intercalated AMs from forming covalent bonds with oxygen, in contrast with AMs on the BLG surface. In the bilayer AM structure, each AM atom forms metallic bonds with its neighboring atoms, which involves the sharing and delocalization of valence electrons among the metal atoms in the bilayer crystal lattice. The AMs form positively charged metal cations and are held together by the delocalized electrons. The presence of the negatively charged electron cloud and its mobility facilitates efficient charge transfer to the outer graphene layers. To quantitatively measure the charge carrier transport from the bilayer AMs to BLG, we utilized an advanced EELS system with a monochromatic electron source offering an energy resolution of 25 meV. Figures 2g−i present the valence loss spectra of the intercalated bilayer AMs in BLG. The black curve represents the original spectra, while the colored curves are the background-subtracted edges of the $C_6M_2C_6$. As a reference, a spectrum of AB stacking BLG without intercalation is included as a grey dashed line in each figure, showing the interband π plasmon peak at 5.06 eV and a smooth absorption background below 4 eV. In the intercalated sample, the interband π plasmon peak is significantly downshifted to 4.76 eV without any interband transition peak[26], indicating complete decoupling of the BLG layers. Additionally, a broad peak in the range of 2 ∼ 2.5 eV is observed for all of the K, Rb, and Cs samples, which can be attributed to the intraband plasmon or so-called charge carrier plasmon[27,28]. This observation is consistent with previous studies on Stage 1 of K intercalated graphite[29]. According to the Drude model, the plasma angular frequency ($\omega_p$) is given by the equation $\omega_p = \sqrt{ne^2/\varepsilon_0 m^*}$, where the $n$ is the carrier density, $e$ is the electron charge, $\varepsilon_0$ is the permittivity of free space, and $m^*$ is the effective mass of electrons. Using this formula, the amount of charge transfer can be calculated to be approximately $1 \sim 1.5 \times 10^{14}$ e⁻/cm².

To further study the charge transfer between the bilayer graphene and intercalated AMs, we performed Raman spectroscopy to the AM intercalated BLG. The Raman spectra of K-, Rb-, and Cs-intercalated BLG are acquired within a sealed glass tube immediately following the intercalation process, ensuring that the samples remain unexposed to air. This setup is depicted in Fig. 3a. The exclusion of exposure to air is crucial for preserving the integrity of the intercalated samples during the Raman measurements. Notably, the Raman G peak of BLG experiences a shift from 1580 cm⁻¹ to 1560 cm⁻¹ following K, Rb, and Cs intercalation as shown in Fig. 3c. The observed shift indicates a remarkably high level of doping, reaching nearly 0.01 - 0.03 e⁻/C atom which is equivalent to $10^{13} \sim 10^{14}$ e⁻/cm² for graphene layers[30–32]. This value is consistent with the estimation derived from the charge carrier plasmon in our EELS data. The extent of n-doping is further corroborated by our transport measurements. This high-level doping will also result in the enhance of electrical conductivity. In terms of transport properties, BLG graphene with prepatterned electrodes was fabricated on a $SiO_2$/Si substrate (Fig. 3b). The $I_d$-$V_g$ characteristics of BLG, as compared to K and Rb intercalated BLG, are illustrated in Fig. 3d. The heavily n-doped intercalated BLG exhibits a charge neutrality point that shifts toward the conduction band, extending beyond the gate voltage of −200V. The sheet resistance of BLG undergoes significant changes upon K and Rb intercalation. Specifically, the sheet resistance transitions from 1303.8 Ω/□ to 306.1 Ω/□ and 367.5 Ω/□ after K and Rb intercalation, respectively. These findings underscore the substantial impact of intercalation on the electronic characteristics of BLG, as evidenced by both Raman spectroscopy and transport measurements.

### Alkali metal intercalation in graphite
Since we have established that bilayer AM structures can form in BLG, an intriguing question arises: can we achieve higher AM storage

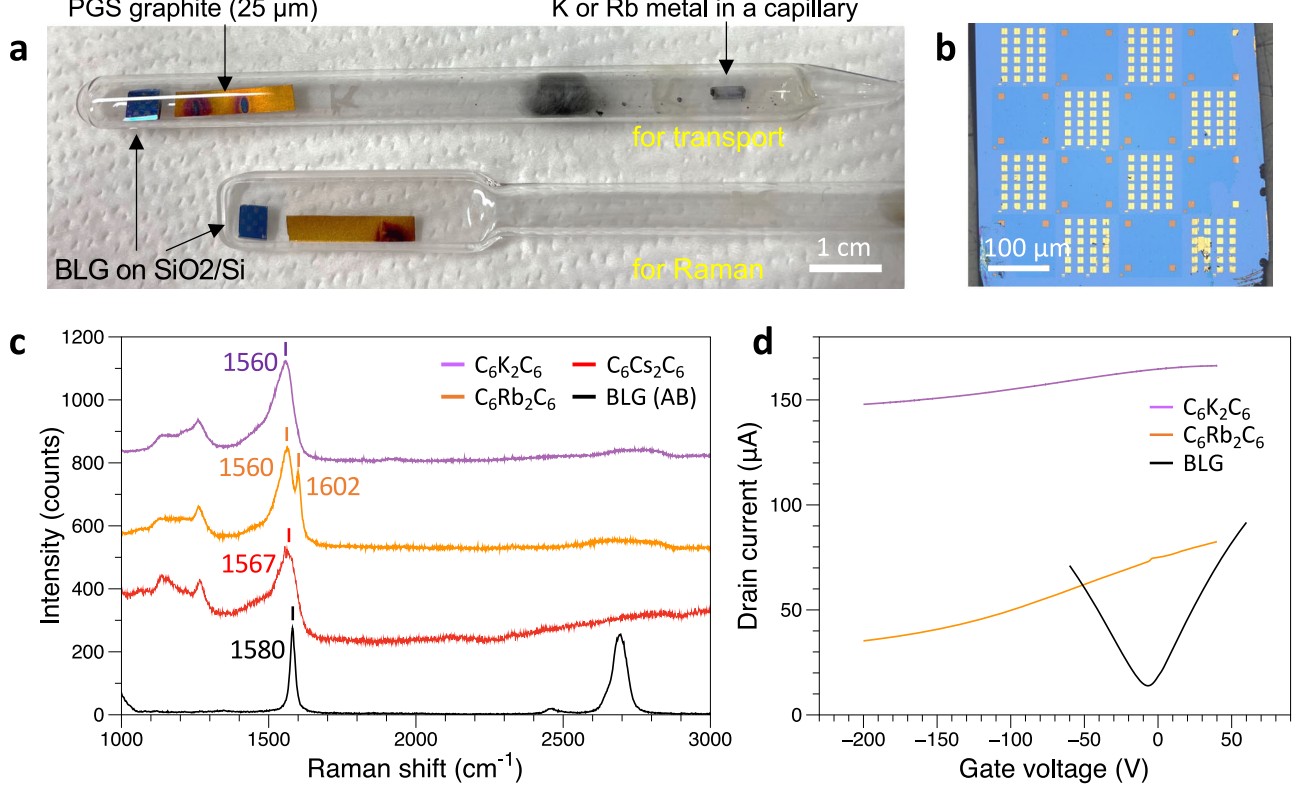

**Fig. 3 | Electronic characteristics of alkali metal (AM) intercalated BLG.**
**a** Photograph of sample prepared by vapor-phase intercalation. **b** Optical image of the BLG with prepatterned Au electrodes on a $SiO_2$/Si substrate for intercalation and electrical transport measurements. **c** Raman spectra of $C_6K_2C_6$, $C_6Rb_2C_6$, $C_6Cs_2C_6$, and BLG. **d** $I_d$-$V_g$ characteristics of $C_6K_2C_6$, $C_6Rb_2C_6$, and BLG measured at room temperature.

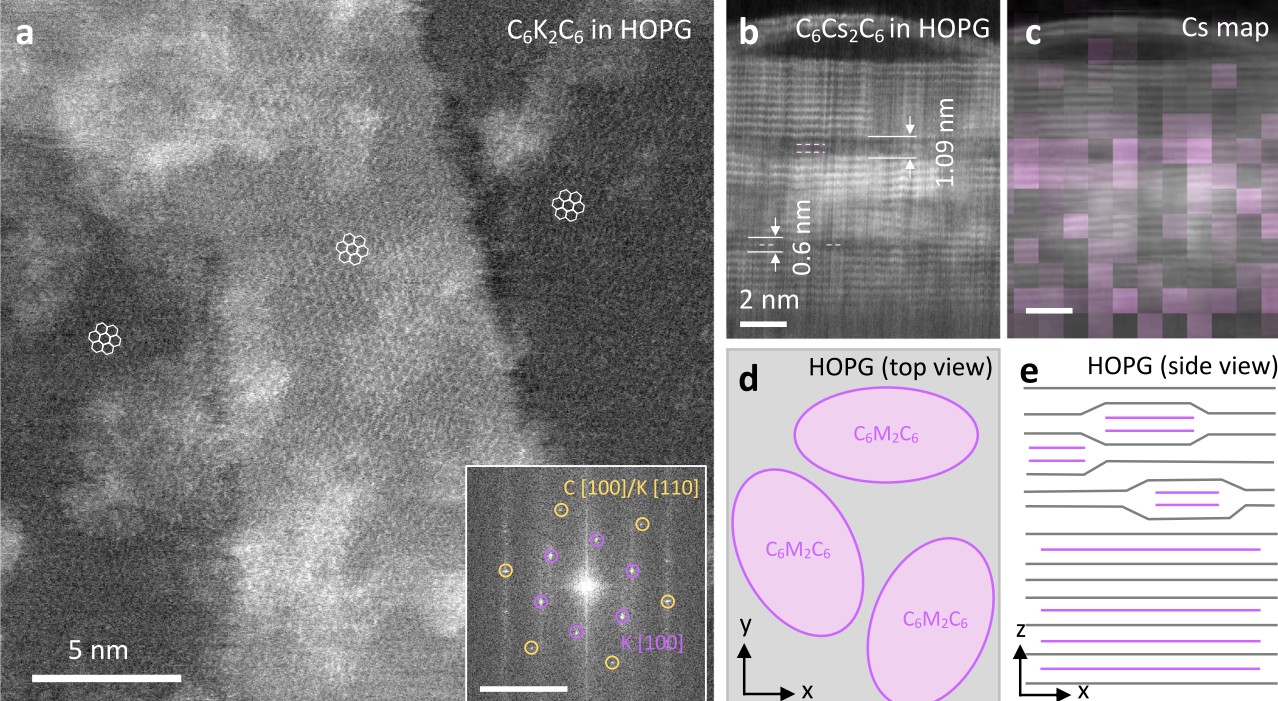

**Fig. 4 | Characterization of K and Cs intercalation in graphite. a** ADF image of K intercalated highly oriented pyrolytic graphite (HOPG) revealing the honeycomb $C_6K_2C_6$ structure in three domains with different numbers of graphene layers. The imaged region has an approximate thickness of 5 nm (Supplementary Fig. 15). The inset displays the corresponding FFT image (scale bar = 5 nm⁻¹). **b** Cross-sectional ADF image of Cs intercalated HOPG. Bilayer Cs is found near the surface layers, while single layer Cs is found in the deeper layers of graphite. **c** Corresponding EELS Cs intensity mapping from the same area as in (**b**). **d** Schematic illustration of the top view of the AM intercalation in HOPG, highlighting the $C_6M_2C_6$ domains in pink. **e** Schematic illustration of the side view of AM intercalation in HOPG, emphasizing the wrapping of the AM bilayer domain by the surface graphite layers and the single-layer AM intercalated in the deeper layer of graphite. The cross-sectional specimen, fabricated by focused ion beam (FIB), has a specimen thickness in y-direction of approximately 50 nm. The darker ADF contrast of the Cs intercalation domain is attributed to its incomplete intercalation within the graphite layers.

capacity by intercalating bilayer AMs into graphite? To investigate this, we prepared exfoliated graphite from highly oriented pyrolytic graphite (HOPG) and performed AM intercalation experiments. Interestingly, we observed that in the thicker graphite films, the majority of the intercalated K and Rb atoms exhibited fast mobility and could not be clearly resolved in STEM observation (see Supplementary Fig. 14 and Movie 1 and 2). However, we found that stable K *hcp* bilayers could be observed in thinner graphite films. Figure 4a displays an ADF image of K-intercalated few-layer HOPG, revealing a honeycomb atomic arrangement similar to that observed in BLG. The thickness of the few-layer HOPG sample where $C_6K_2C_6$ structure was observed was approximately 5 nm (Supplementary Fig. 15). We hypothesize that these relatively unstable structures correspond to the intercalation of single-layer AM situated in the interior of the thick graphite layers. In contrast, the surface layers of graphite demonstrate the ability to accommodate bilayer AM intercalation, which tends to be more stable under e-beam irradiation. Consequently, we conclude that the honeycomb structure depicted in Fig. 4a represents an overlapped configuration, comprising AM bilayers in the top few layers of graphite and single K layers within different graphene sheet beneath. The absence of visualizing bilayer AMs in thick graphite may be attributed to the fact that these bilayer AMs near the surface layers are out of focus in our STEM imaging when viewed from the top. The presence of a substrate could potentially limit the expansion of graphene layers on the side attached to the substrate. However, for both BLG and the top surface of HOPG, the expansion of interlayer spacing remains effective and is not significantly influenced by the substrate allowing AM bilayer intercalation.

To further investigate the bilayer structure of intercalated AM from a side view, we performed cross-sectional STEM imaging on Cs

intercalated HOPG, as shown in Fig. 4b. Additionally, Fig. 4c presents an EELS mapping overlaid onto the ADF cross-sectional image, highlighting the Cs intensity distribution. The Cs signal was found to be widely distributed within the graphite layers, with a distinct dark contrast layer displaying a significant Cs signal. Furthermore, a two-layer Cs intercalated structure was observed within the dark contrast region expanding the interlayer spacing between the graphite layers to 1.09 nm, consistent with the model obtained from our DFT calculations (Fig. 1e and Supplementary Fig. 7). Additionally, single-layer Cs intercalated structures were also visualized in the deeper layers of HOPG, with the graphite interlayer spacing being expanded to 0.6 nm. The darker contrast of Cs in the cross-sectional image can be attributed to incomplete intercalation within the graphite layers. In our bilayer graphene intercalation study, it is worth noting that the AM bilayer intercalants do not fully occupy the interlayer gap in BLG (Supplementary Fig. 10). This could be due to some deintercalation during specimen preparation or STEM observation. The void observed in the upper part of Fig. 4b is likely a consequence of surface oxidation in the Cs-intercalated HOPG during the sample preparation for cross-sectional imaging (Supplementary Fig. 16).

Schematic illustrations of the intercalation structure are provided in Figs. 4d–e. The pink-colored regions represent domains with the well-ordered $C_6M_2C_6$ structures, while the remaining regions may also contain intercalated AMs but with greater mobility under the electron beam, potentially indicating single-layer AM intercalation areas situated in the middle of the thick graphite layers. From the side view, the small AM bilayer domains are surrounded by the graphite layers, with the cross-sectional specimen prepared using focused ion beam (FIB) having a thickness of approximately 50 nm. As a result, even for heavy AMs such as Cs, the small Cs intercalation domains still exhibit darker

ADF contrast compared to the surrounding graphite layers. Theoretical calculations have estimated that the charge capacity of BLG can exceed the theoretical value by inserting an additional layer of AM atoms[13]. Our discovery proves that the AM bilayer can intercalate in few-layer graphite and suggests a possible way to improve the battery capacity.

In conclusion, our study has demonstrated the formation of stable bilayer AM structures through intercalation in BLG. While the presence of bilayer AM structures can also be observed in multilayer graphite, it is typically limited to small domain sizes or more easily observed in few-layer graphene (FLG). Interestingly, despite utilizing high-resolution imaging techniques, we did not obtain a clear atomic image of the single-layer form of AM intercalation in our experiment. This suggests that the single-layer AM structure may be highly mobile and less stable under electron beam irradiation. The interlayer spacing in thick graphite is less flexible compared to BLG and FLG, leading the AM bilayer to predominantly intercalate into the surface layers of graphite. This implies that intercalating AM atoms into graphite layers with lower crystallinity or reduced thickness could potentially enhance the AM storage capacity. Our findings not only contribute insights into the intercalation of AMs in BLG but also provide valuable information about the potential behavior of AM ions during electrolyte intercalation.

## Methods

### Material growth and alkali metal intercalation
Isolated single crystal BLG domains were grown on Cu foil by ambient pressure chemical vapor deposition (CVD)[15]. The Cu foil was heated in a tube furnace up to 1050 °C for 40 min with a constant flows of Ar (300 sccm) Ar and $H_2$ (10 sccm). The sample was annealed for 90 min, then methane (80 ppm) mixed with the flows of Ar (300 sccm) and $H_2$ (15 sccm) was fed into the reaction chamber for 7 min to form BLG. After the growth, the Cu foil was moved to the cooling zone under the flow of Ar and $H_2$.

Another uniform BLG sheet was synthesized by ambient pressure CVD using a Cu–Ni alloy thin film deposited on c-plane sapphire[25]. The Ni and Cu thin film was deposited by radio frequency (RF) sputtering. The substrate was set at -500 °C and 80 °C for Ni and Cu deposition. The Cu–Ni film was alloyed by heating at 950 °C in a Ar/$H_2$ (Ar: 195 sccm, $H_2$: 5 sccm) flow for 1 h at ambient pressure. Then $CH_4$ (200 ppm) feedstock as supplied for 10 min to 1 h at temperature ramped to 1085 °C. After growth, the sample was immediately moved to the cooling zone to room temperature.

The BLG sheet was transferred onto a TEM grid by etching the Cu foil by diluted HCl or etching a Cu–Ni film with an aqueous ammonium persulfate etching solution, followed by a well-developed clean transfer process[33,34]. After transference to the TEM grid, the BLG was vacuum sealed in a reaction glass ampoule for the intercalation of K, Rb, and Cs. The vacuum reaction tube was opened in an Ar atmosphere glove box and the AM@BLG specimen was transferred to the TEM using a JEOL vacuum transfer holder without exposure to air.

### K and Rb intercalation
The intercalated specimens were synthesized by the vapor method. The alkali metals K (Sigma Aldrich 99.95%) and Rb (Sigma Aldrich 99.95%) were used without further purification. The alkali metals were put into a glass ampoule with the BLG and PGS graphite sheet (EYGS, Panasonic Co.), where the molar ratio of K/C and Rb/C was adjusted to be 8 under an Ar atmosphere. The glass ampoules were sealed after evacuating and heated to 200 °C for a few days. The reaction endpoints were confirmed by the coloration to characteristic golden colors of the PGS specimens[35].

### Cs intercalation
The intercalation of Cs into the graphite specimen was carried out according to the following method. At first $CsC_8$ was prepared from small pieces of Grafoil sheet (GrafTech Co., GTA grade). Metallic Cs (Sigma Aldrich 99.95%) and small pieces of Grafoil (typically 15 mm × 5 mm × 0.4 mm) were put in a glass ampoule under vacuum where the molar ration of C/Cs was adjusted to be 8. The glass ampoule was then set in an oven held at 230 °C for a week. The formation of $CsC_8$ was confirmed by the coloration to characteristic brownish golden[35].

### STEM and EELS
STEM images were acquired by using JEOL-tripleC#3, an ultrahigh vacuum JEOL-ARM200F-based microscope equipped with a JEOL delta corrector and a cold field-emission gun operating at 60 kV. The probe current was approximately 15 pA for intercalated Cs and 5 pA for K and Rb, and the convergence semi-angle and the inner acquisition semi-angle were 37 and 76 mrad, respectively. A typical ADF image has a resolution of 1024 × 1024 pixels and was captured using a 38.5 μs pixel time. EELS core-loss spectra were obtained using a Gatan Rio CMOS camera optimized for low-voltage operation. EEL spectra were acquired by line scanning with an exposure time of 0.1 s/pixel. EELS low loss spectra were performed by using JEOL-tripleC#2, a low-voltage thermal 60 kV microscope equipped with a double Wien-filter monochromator and delta correctors for TEM and STEM. A convergence semi-angle and the inner acquisition semi-angle were 43 mrad and 125 mrad. The probe current was 8 pA with an energy resolution of 45 meV after inserting a 0.5 μm slit. All STEM images and EELS were collected at room temperature.

### Raman characterization
Raman spectra were acquired by using a JASCO NRS-5500 spectrometer, which was equipped with a 5× objective lens and a DU420_OE CCD detector with 1024 channels cooled to −70 °C. The excitation laser had a wavelength of 532 nm and a power of 5.9 mW. The total exposure time for the measurements was 100 s.

### Electrical transport measurement
For the electrical characterization of the intercalated BLG, the as-grown BLG was initially transferred onto a $SiO_2$/Si substrate. Subsequently, the BLG was patterned into squares with 1 mm sides for sheet resistance measurements, and into stripes of 20 μm width to serve as channels for the field-effect transistors (FETs). Electrodes were then fabricated through the deposition of a 50 nm layer of Au. Following this, the samples underwent the intercalation process. To prevent exposure to ambient conditions, a custom-made sample handler was employed to transport the intercalated samples into the chamber of a device probe station fabricated by Thermal Block Co. The electrical measurements were finally conducted at room temperature and at low pressure (-$10^{-2}$ Pa) using a Keysight B1500A semiconductor device parameter analyzer

### Density functional theory calculations
The geometries of K, Rb, and Cs intercalation in BLG were optimized using density functional theory (DFT) implemented in the CASTEP module of the Materials Studio ver. 7.0 software (Accelrys Co.). Structures were generally obtained using ultra-fine quality gradient approximation (GGA) – PBE functionals[36]. The calculation parameters for ultra-fine quality are: the convergence threshold for the maximum change in energy, maximum force, maximum stress and maximum displacement, which were set to $5.0 \times 10^{-7}$ eV/atom, 0.01 eV/Å, 0.02 GPa, and $5.0 \times 10^{-4}$ Å, respectively. An SCF tolerance smaller than $5.0 \times 10^{-7}$ eV/atom was regarded as converged. We also used a 390.0 eV

energy cutoff and a $1 \times 1 \times 1$ K-point set for the band structure and DOS calculation.

## Data availability

All data supporting the findings of this article and its Supplementary Information will be made available upon request to the authors.

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

## Acknowledgements

This work was financially supported by the JSPS Grant-in-Aid for Scientific Research on Innovative Areas "Science of 2.5 Dimensional Materials: Paradigm Shift of Materials Science Toward Future Social Innovation". Y.C.L. acknowledge to JSPS-KAKENHI (18K14119, 22H05478). K.S. acknowledge to JSPS-KAKENHI (16H06333, 21H05235, 22F22358), the JST-CREST program (JPMJCR20B1, JMJCR20B5, JPMJCR1993), ERC "MORE-TEM" (951215), and the JSPS A3 Foresight Program. H.A. acknowledge to JSPS-KAKENHI (18H03864, 21H05232, 21H05233, 23K18878). P.W.C. acknowledge to the Ministry of Science and Technology (MOST) Taiwan Grants MOST 109-2124-M-007-002-MY3, MOST 109-2112-M-007-027-MY3, MOST 106-2628-M-007-003-MY3, and MOST 109-2124-M-006-001 as well as Academia Sinica (AS) Grant AS-TP–106-A07. We acknowledge Prof. Noboru Akuzawa for conducting the Cs and ethylene intercalation experiments.

## Author contributions

Y.C.L., R.M., H.A., and K.S. designed the project. P.S.F., M.D.S., H.A., and P.W.C. synthesized BLG. R.M. performed the AM intercalation experiment. Y.C.L. and Q.L. performed the STEM experiments. P.S.F and H.A. fabricated the devices and performed the electrical transport measurements. Y.C.L. analyzed the data. Y.C.L. and K.S. co-wrote the paper. All authors commented on the manuscript.

## Competing interests
The authors declare no competing interests.
