## [Peer Review File · Nature Communications]

Alkali Metal Bilayer Intercalation in GrapheneREVIEWER COMMENTS

Reviewer #1 (Remarks to the Author):

The author presented a work which investigated the structure of K, Cs, Ru intercalated bilayer graphene. Mainly the STEM and EELS were used to understand the formed structure. Although, the formed bilayer structure is well-characterized and discussed, the work lacks the connection with the application, as compare with previous reported similar works for example the intercalation of Li, Na and K for battery. This reviewer cannot find a significant novelty and originality of this work.

The intercalation process of the K, Cs, Ru need to be clarified, especially the edge intercalation or the defect (if polycrystalline graphene) intercalation. In addition, the different intercalation stage should be characterized, for example the difference between one layer intercalation and two layers intercalation. Moreover, how to achieve and evaluate if the sample (bilayer graphene) was full interaction? Could the substrate influence the structure of intercalated Cs? if the bilayer graphene are deposited on the different substrates.

The sample was prepared by the vapor phase intercalation. However, the ions intercalation into graphene/graphite for energy storage are usually governed by a different driven force. The aim to understand the alkali interaction as the authors introduced is mainly for energy storage and ionic application. Therefore, the intercalated graphene prepared by different driven force need to be investigated.

Why does the intercalation of second layer Cs cause much larger interlayer distance increase as compare with the first layer intercalation?

As claimed by the author that there is charge transfer between the bilayer graphene and intercalated AMs. This need to be characterized and verified by Raman as well.

For the intercalation into graphite as compare with the bilayer graphene, the author claimed that the full intercalation was not observed. Could the AMs absorbed on the top surface of sample affect the intercalation of AMs into the bilayer graphene or graphite, and thus the intercalated structure?

For the potassium ion battery, it has been demonstrated that the potassium ions can be reversibly intercalated into graphite with the formation of uniform intercalated structure. Please discuss more about why the monolayer intercalation was formed in battery but bilayer intercalation was formed in this work.

Reviewer #2 (Remarks to the Author):

In the manuscript "Alkali Metal Bilayer Intercalation in Graphene", the author use STEM to directly visualize the atomic structure of intercalated alkali metals in bilayer and few-layer graphene, adopt EELS to measure the charge transfer between alkali metals and graphene layers, the reveal the intercalation mechanism. This is a novel fundamental work. It could be considered further for publication in Nature Communications after major revisions.

1. In the abstract, the author declared that "this research provides the first visualization of the precise atomic structure involved in the over-doped alkali metal intercalation." Is it contradict with Ref.12 that utilizes TEM to image multilayer stacking Li within bilayer graphene.
2. The author declared that no single-layer, three-layer, or four-layer arrangements of AMs layer were observed. What the image of 1-4 layers should be? How to distinguish between 2 layers and 4 layers from the image? If 2 layers is confirmed by EELS quantification, it is better to present the result of K and Rb.
3. In the manuscript, the author calculates the amount of charge transfer, it is better to present more details about the estimation.
4. How to confirm that the honeycomb structure in Figure 3a is bilayer of K. As the author declared that the cross-sectional image of intercalated HOPG may contain single-layer AM intercalation area, is it possible that the honeycomb image resulted from the overlap of single K layers within different graphene sheets.
5. What does the void in the upper part of Figure 3b? Does it result from the intercalation of Cs and deintercalation during specimen preparation?
6. Some errors in the manuscripts. It seems that no author is employed by affiliation 6; "4.3 Gpa" should be "4.3 GPa"; "1~1.5x10¹⁴" should be "1~1.5×10¹⁴".

Reviewer #3 (Remarks to the Author):

Lin et al. report a scanning transmission electron microscopy (STEM) and electron energy loss spectroscopy (EELS) study of alkali-metal intercalated bi- and few-layer graphene samples. Their major new finding is the observation of stable alkali metal (AM) bilayer crystals that occur in these very thin forms of graphitic carbon but not in bulk graphite. Furthermore, the authors report stable bilayers of several AMs: cesium, potassium, and rubidium. The work is of high quality and the experimental findings constitute an important advance over previous work in the field, pointing to possible knowledge gaps in the understanding of intercalation chemistry of graphitic samples at the limit of ultrathin crystals. The phenomenology could potentially further be generalizable to the intercalation of thin specimen of other 2D materials and may have implications for the design of ion insertion electrodes. Therefore, it is my opinion that the manuscript presented by Lin et al. is an excellent fit for Nature Communications and may attract attention from a broad audience, including but not limited to the transmission electron microscopy, 2d materials, and electrochemical energy storage communities.

Overall, I find the manuscript well written and the presentation of the results well suited for publication in Nature Communications. I was nonetheless confused about a few things I encourage the authors to address to make the manuscript more accessible and comprehensible in particular for non-experts in scanning transmission electron microscopy.

- 1) How did the authors determine the bilayer nature of the graphene samples studied based on their data? Specifically, how did the authors rule out the samples studied were trilayer graphene where AM single layers could sit in neighboring interlayer spaces?
- 2) How did the authors determine the AM bilayers studied were confined in the interlayer gap between two layers of graphene, rather than located on top of or below these based on their data?
- 3) I am not an expert in ADF imaging, and was confused by the presentation of the data in Fig. 1. I expected to first of all see imaging/diffraction evidence for bilayer graphene without Cs, only then to be introduced to C6Cs2C6 in a second step. Instead, Figure 1 is all about C6Cs2C6 and the reader is left to deduce that for some reason the carbon atoms are not visible, instead the contrast mainly stems from Cs, the diffraction pattern itself would be different for bilayer graphene without Cs, etc. I recommend the authors to introduce what is known first, which could include merging Extended Data Fig. 3 with Fig. 1 for example.
- 4) Regarding Extended Data Fig. 3: Did the authors determine the 2.7 degree twist angle between graphene layers from the diffraction pattern? If so, is there also a 2.7 degree twist in the C6Cs2C6 region? Doesn't the $(\sqrt{3} \times \sqrt{3}) R 30 \text{ deg}$ structure necessitate AA stacking of the graphene layers or does that supercell refer to only one of the two graphene layers?
- 5) Could the authors expand their discussion by commenting on what they think is special about the 4 nm thickness beyond which the authors assert no more bilayer AMs would occur in graphite? Couldn't bilayer AMs still occur near the surface of an even thicker graphite flake, but maybe not near its center?
- 6) Could the authors expand their discussion by commenting on why they think only bilayers stable? What do the authors think is the mechanism that rules out the formation of trilayers or thicker AM layers in their experiment once the two graphene sheets have been separated?

Minor comments:

- 1) Line 192: "charge carrier plasmon from the bilayer AMs to BLG" may have to read "charge carrier transport from the bilayer AMs to BLG" instead.
- 2) Legends in Fig. 2 g-l all read "C6K2C6" which needs correction.
- 3) Illegibly small labels/axis numbers in Figs. 2 d-f (insets),

REVIEWER COMMENTS

Reviewer #1 (Remarks to the Author):

The author presented a work which investigated the structure of K, Cs, Ru intercalated bilayer graphene. Mainly the STEM and EELS were used to understand the formed structure. Although, the formed bilayer structure is well-characterized and discussed, the work lacks the connection with the application, as compare with previous reported similar works for example the intercalation of Li, Na and K for battery. This reviewer cannot find a significant novelty and originality of this work.

The intercalation process of the K, Cs, Ru need to be clarified, especially the edge intercalation or the defect (if polycrystalline graphene) intercalation. In addition, the different intercalation stage should be characterized, for example the difference between one layer intercalation and two layers intercalation. Moreover, how to achieve and evaluate if the sample (bilayer graphene) was full interaction? Could the substate influence the structure of intercalated Cs? if the bilayer graphene are deposited on the different substates.

Response: We appreciate the reviewer's comment. In our experimental approach, we utilized both single crystal BLG (ACS Nano 7, 2587-2594 (2013)) and polycrystalline BLG (Chem. Mater. 28, 4583-4592 (2016)) for AM intercalation, and observed a $C_6M_2C_6$ intercalated structure in both cases. Consequently, both edge and defect intercalation processes should presumably contribute to the overall intercalation of AM in BLG.

As outlined in the method, a large pyrolytic graphite sheet (PGS) measuring $0.5\text{ cm} \times 2\text{ cm} \times 25\text{ }\mu\text{m}$ was placed in a glass ampoule alongside the BLG sample on a TEM grid (0.3 cm in diameter). The PGS serves as an indicator for monitoring the intercalation stage. When the PGS turns a golden color, it signifies that the intercalation has reached stage 1 for the graphite, marking the endpoint of the reaction. Given that the BLG sample shares the same environment as the PGS, we infer that the relatively smaller size of the BLG sample, compared to the PGS, should also meet the conditions for stage 1 full intercalation, as observed with the PGS.

Furthermore, we placed the BLG or exfoliated HOPG on a SiO_2/Si substrate for AM intercalation. Cross-sectional STEM imaging revealed a bilayer form of Cs intercalated in HOPG. The reviewer's concern is valid; the presence of a substrate could potentially limit the expansion of graphene layers on the side attached to the substrate. However,

for both BLG and the top surface of HOPG, we believe that the expansion of interlayer spacing remains effective and is not significantly influenced by the substrate allowing AM bilayer intercalation. The corresponding comments are addressed to the revised manuscript.

The sample was prepared by the vapor phase intercalation. However, the ions intercalation into graphene/graphite for energy storage are usually governed by a different driven force. The aim to understand the alkali interaction as the authors introduced is mainly for energy storage and ionic application. Therefore, the intercalated graphene prepared by different driven force need to be investigated.

Response: We appreciate the insightful comment from the reviewer. Indeed, we acknowledge that the driving forces for vapor phase intercalation and for electrochemical processes should differ. It is crucial to highlight that vapor phase intercalation represents a relatively “clean” process compared to electrolyte intercalation. Consequently, we can achieve a clear visualization of the atomic structure of AM in BLG. Our findings also offer valuable insights into the potential behavior of AM ions during electrolyte intercalation.

We want to emphasize that the electrochemical intercalation process in TEM constitutes a distinct full study and challenging experiment setup. We are actively engaged in this project, and the results will be presented and discussed separately. In our manuscript, we have rephrased the relevant sections and included comments to underscore this point.

Why does the intercalation of second layer Cs cause much larger interlayer distance increase as compare with the first layer intercalation?

Response: We appreciate the reviewer’s observation, and we would like to provide additional insights based on our DFT calculations. According to our findings, the interlayer distance of BLG is calculated to be 5.95Å with a single layer of Cs intercalation, expanding to 11.25Å with two layers of Cs intercalation (**Supplementary Fig. 7**). In the atomic model of Cs crystal at high-pressure phase (**Supplementary Fig. 8**), the distance between two Cs layers along the fcc(111) plane is approximately 3.45Å.

The rationale behind the further expansion of the BLG interlayer distance with two Cs layers of intercalation can be attributed to the repulsive forces between Cs atoms. Cs metal atoms tend to undergo electron charge transfer to the outer graphene layers, resulting in a partial positive charge on each Cs atomic layer. Consequently, this leads

to an enlargement of the distance between the two Cs layers to 5.02Å due to the repulsive electrostatic forces involved. We have included these explanations in the revised manuscript to provide a clearer understanding of our DFT calculations and the factors influencing the interlayer distances in BLG with Cs intercalation.

As claimed by the author that there is charge transfer between the bilayer graphene and intercalated AMs. This need to be characterized and verified by Raman as well.

Response: The Raman spectra of K-, Rb-, and Cs-intercalated BLG are acquired within a sealed glass tube immediately following the intercalation process, ensuring that the samples remain untouched by air. This setup is depicted in Figure 3a. The exclusion of exposure to air is crucial for preserving the integrity of the intercalated samples during the Raman measurements. Notably, the Raman G peak experiences a shift from 1580 cm^{-1} to 1560 cm^{-1} following K and Rb intercalation. The observed shift indicates a remarkably high level of doping, reaching nearly 0.01~0.03 e^-/C atom which is equivalent to $10^{13}\sim 10^{14}$ e^-/cm^2 for graphene layers (PRB 84, 241404 (2011)). This value is consistent with the estimation derived from the charge carrier plasmon in our EELS data. The extent of n-doping is further corroborated by our transport measurements.

In terms of transport properties, the I_d - V_g characteristics of BLG, as compared to K and Rb intercalated BLG, are illustrated in **Figure 3d**. The heavily n-doped BLG exhibits a charge neutrality point that shifts toward the conduction band, extending beyond the gate voltage of -200V. The sheet resistance of BLG undergoes significant changes upon K and Rb intercalation. Specifically, the sheet resistance transitions from 1303.8 ohm/sq to 306.1 ohm/sq and 367.5 ohm/sq after K and Rb intercalation, respectively. These findings underscore the substantial impact of intercalation on the electronic characteristics of BLG, as evidenced by both Raman spectroscopy and transport measurements.

Figure 3 | Electronic characteristics of AM-intercalated BLG. (a) Photograph of sample prepared by vapor-phase intercalation. (b) Optical image of the BLG with prepatterned Au electrodes on a SiO₂/Si substrate for intercalation and electrical transport measurements. (c) Raman spectra of C₆K₂C₆, C₆Rb₂C₆, C₆Cs₂C₆, and BLG. (d) Id-V_g characteristics of C₆K₂C₆, C₆Rb₂C₆, and BLG measured at room temperature.

For the intercalation into graphite as compare with the bilayer graphene, the author claimed that the full intercalation was not observed. Could the AMs absorbed on the top surface of sample affect the intercalation of AMs into the bilayer graphene or graphite, and thus the intercalated structure?

Response: Surface absorption is an inherent aspect of the gas phase intercalation sample. However, even when the BLG is coated with amorphous or oxidized AMs, the intercalated structure maintains the consistent bilayer form of the C₆M₂C₆ structure. In **Supplementary Figure 13b**, an ADF image of Rb-intercalated BLG with surface deposition is presented, revealing the crystalline structure of Rb in BLG. However, due to the presence of a surface amorphous layer, the imaging quality is somewhat diminished (blurred FFT spots of Rb in BLG).

To address this, we subjected the sample to an e-beam shower for 5 min over a large area (~ 100 μm²). This treatment effectively removed surface-deposited AMs or their oxides, as depicted in **Supplementary Figure 13a**. Importantly, the crystalline intercalated AMs remained and were clearly visualized in **Supplementary Figure 13c**.

Based on these observations, we conclude that surface deposition does not adversely affect the intercalated structure. This insight has been incorporated into the revised manuscript for a more comprehensive discussion of the impact of surface adsorption on the gas-phase intercalation process.

Supplementary Fig. 13 | Removal of surface deposited amorphous AM by e-beam. (a) Low-magnification TEM image of the Rb-intercalated BLG sample. The area to the right side from the yellow dash line depicts the region exposed to the e-beam shower, resulting the removal of most surface-deposited materials. (b) ADF image captured from the non-e-beam showered region, where the crystalline Rb structure is obscured by surface contamination. (c) ADF image obtained from the e-beam showered region, revealing a clear resolution of the crystalline intercalated Rb structure.

For the potassium ion battery, it has been demonstrated that the potassium ions can be reversibly intercalated into graphite with the formation of uniform intercalated structure. Please discuss more about why the monolayer intercalation was formed in battery but bilayer intercalation was formed in this work.

Response: In our STEM studies, we observed the rapid removal of layered structures

by the e-beam in thick HOPG (**Supplementary Figure 14 and Movie 1 and 2**). These highly mobile intercalated AMs potentially represent the single-layer AM intercalation, dominating in the middle of thick graphite. However, due to the relatively instability of these single-layer AM layers, obtaining a clear atomic image from them is challenging. In contrast, the surface layers of graphite demonstrate the ability to accommodate bilayer AM intercalation, which proves more stable under e-beam irradiation and can be visualized clearly.

Upon careful reanalysis of the cross-sectional STEM image, we found possible evidence of single-layer Cs in the deeper layers of HOPG, as shown in **Figure 4b**. The honeycomb structure depicted in **Figure 4a** represents an overlapped configuration, comprising AM bilayers in the top few layers of graphite and single K layers within different graphene sheet beneath. Therefore, we emphasize that both single-layer AM and bilayer AM are visualized in our work. While we are able to observe the AM bilayer in thin HOPG, the absence of visualizing bilayer AMs in thick graphite may be attributed to the fact that these bilayer AMs near the surface layers are out of focus in our STEM imaging when viewed from the top. The related discussion has been added to the revised manuscript and Figure 4 has been updated.

Supplementary Fig. 14 | STEM characterization of the K and Rb intercalated HOPG. (a-c) Sequential ADF images of K-intercalated HOPG extracted from Movie 1. (d-f) Sequential ADF images of Rb-intercalated HOPG extracted from Movie 2. In both samples, we observe the removal of the layered structure due to e-beam scanning,

suggesting the presence of mobile layered structures, likely corresponding to single-layer K and Rb intercalated within the graphite layers.

Figure 4 | Characterization of K and Cs intercalation in graphite. **a**, ADF image of K intercalated HOPG revealing the honeycomb $C_6K_2C_6$ structure in three domains with different numbers of graphene layers. The imaged region has an approximate thickness of 5 nm (**Supplementary Figure 15**). **b**, Cross-sectional ADF image of Cs intercalated HOPG. Bilayer Cs is found near the surface layers, while single layer Cs is found in the deeper layers of graphite. **c**, Corresponding EELS Cs intensity mapping from the same area as in **b**. **d**, Schematic illustration of the top view of the AM intercalation in HOPG, highlighting the $C_6M_2C_6$ domains in pink. **e**, Schematic illustration of the side view of AM intercalation in HOPG, emphasizing the wrapping of the AM bilayer domain by the surface graphite layers and the single-layer AM intercalated in the deeper layer of graphite. The cross-sectional specimen, fabricated by FIB, has a specimen thickness in y-direction of approximately 50 nm. The darker ADF contrast of the Cs intercalation domain is attributed to its incomplete intercalation within the graphite layers.

Reviewer #2 (Remarks to the Author):

In the manuscript "Alkali Metal Bilayer Intercalation in Graphene", the author use STEM to directly visualize the atomic structure of intercalated alkali metals in bilayer and few-layer graphene, adopt EELS to measure the charge transfer between alkali metals and graphene layers, the reveal the intercalation mechanism. This is a novel

fundamental work. It could be considered further for publication in Nature Communications after major revisions.

We are grateful to the reviewer for the positive comment.

1. In the abstract, the author declared that “this research provides the first visualization of the precise atomic structure involved in the over-doped alkali metal intercalation.” Is it contradict with Ref.12 that utilizes TEM to image multilayer stacking Li within bilayer graphene.

Response: We thank the referee for bringing this to our attention. Indeed, Ref.12 is the first report that attempts to visualize Li intercalation in BLG by TEM. In the revised manuscript, we have removed the term “first” from the sentence.

2. The author declared that no single-layer, three-layer, or four-layer arrangements of AMs layer were observed. What the image of 1-4 layers should be? How to distinguish between 2 layers and 4 layers from the image? If 2 layers is confirmed by EELS quantification, it is better to present the result of K and Rb.

Response: We have done the EELS quantification of K and Rb, too. The spectra confirm again the chemical compositions of $C_6K_2C_6$ structure $C_6Rb_2C_6$. We additionally conducted STEM image simulations for 1-4 layers of AM, as illustrated in **Supplementary Fig. 11**. Through the quantitative ADF contrast comparison, we can effectively distinguish between 2 layers from 3 and 4 layers. The results of EELS quantification for K and Rb, confirming the $C_6M_2C_6$ structure, are summarized in **Supplementary Fig. 12**. Based on our findings, we can confidently conclude that the intercalated AM consistently adopts a bilayer form.

Supplementary Fig. 11 | Structural comparison of intercalated Cs in multilayer form. (a) STEM simulation and cross-section atomic model for 1-4 layers of Cs with a $(\sqrt{3} \times \sqrt{3})R30^\circ$ structure. (b) STEM simulation for the stacking of two Cs bilayers with a twist angle of 10° .

Supplementary Fig. 12 | EELS quantification for the $C_6K_2C_6$ and $C_6Rb_2C_6$. (a) In $C_6K_2C_6$, the calculated atomic ratio for C is 82.81 at%, and for K, it is 17.19 at%. The K:C ratio is slightly higher than the expected value (1:6) due to the overlapping of the K L-edge and the C K-edge, resulting in a higher background signal for K. (b) In $C_6Rb_2C_6$, the calculated atomic ratio for C is 87.21 at%, and for Rb, it is 12.79 at%, which is consistent with the $C_6Rb_2C_6$ structure.

3. In the manuscript, the author calculates the amount of charge transfer, it is better to present more details about the estimation.

Response: We thank the reviewer for pointing out. We have added the following

information to the revised manuscript:

According to the Drude model, the plasma angular frequency (ω_p) is given by the equation $\omega_p = \sqrt{ne^2/\epsilon_0 m^*}$, where the n is the carrier density, e is the electron charge, ϵ_0 is the permittivity of free space, and m^* is the effective mass of electrons. The amount of charge transfer can be calculated using this formula.

4. How to confirm that the honeycomb structure in Figure 3a is bilayer of K. As the author declared that the cross-sectional image of intercalated HOPG may contain single-layer AM intercalation area, is it possible that the honeycomb image resulted from the overlap of single K layers within different graphene sheets.

Response: We appreciate the reviewer for bringing this to our attention. Based on our observations in thick graphite, we notice that the intercalated K layers exhibit significant mobility under e-beam irradiation. We hypothesize that these relatively unstable structures correspond to the intercalation of single-layer AM situated in the middle of the thick graphite layers. In contrast, the surface layers of graphite demonstrate the ability to accommodate bilayer AM intercalation, which tends to be more stable under e-beam irradiation.

Consequently, we conclude that the honeycomb structure depicted in **Figure 4a** represents an overlapped configuration, comprising AM bilayers in the top few layers of graphite and single K layers within different graphene sheet beneath. This clarification has been incorporated into the revised manuscript. The schematic illustration for this model in **Figure 4e** is also updated.

5. What does the void in the upper part of Figure 3b? Does it result from the intercalation of Cs and deintercalation during specimen preparation?

Response: The void observed in the upper part of Figure 3b is likely a consequence of surface oxidation in the Cs-intercalated HOPG during the sample preparation for cross-sectional imaging. In **Supplementary Figure 16**, the results of EELS color mapping for oxygen in the Cs-intercalated HOPG are presented. The green-colored pixels indicate the presence of oxygen signals accumulating at the sample surface. Additionally, some oxygen signals can also be detected on the sliced surface, which is highly likely to occur during the sample preparation process.

Supplementary Fig. 16 | EELS characterization to the oxidation behavior in the cross-sectional sample. (a) EELS colored mapping for Cs-intercalated HOPG. The green pixels represent oxygen intensity mapping. The sample surface contains a high density of oxygen, and some oxygen is also present in the cross-sectional plane, likely due to the sample preparation process. (b) EELS profile extracted from the red box in (a).

6. Some errors in the manuscripts. It seems that no author is employed by affiliation 6; “4.3 Gpa” should be “4.3 GPa”; “ $1\sim 1.5 \times 10^{14}$ ” should be “ $1\sim 1.5 \times 10^{14}$ ”.

Response: We thank the reviewer’s through examination of our manuscript. The errors highlighted have been rectified in the revised manuscript.

Reviewer #3 (Remarks to the Author):

Lin et al. report a scanning transmission electron microscopy (STEM) and electron energy loss spectroscopy (EELS) study of alkali-metal intercalated bi- and few-layer graphene samples. Their major new finding is the observation of stable alkali metal (AM) bilayer crystals that occur in these very thin forms of graphitic carbon but not in bulk graphite. Furthermore, the authors report stable bilayers of several AMs: cesium, potassium, and rubidium. The work is of high quality and the experimental findings constitute an important advance over previous work in the field, pointing to possible knowledge gaps in the understanding of intercalation chemistry of graphitic samples at the limit of ultrathin crystals. The phenomenology could potentially further be generalizable to the intercalation of thin specimen of other 2D materials and may have implications for the design of ion insertion electrodes. Therefore, it is my opinion that the manuscript presented by Lin et al. is an excellent fit for Nature Communications and may attract attention from a broad audience, including but not limited to the

transmission electron microscopy, 2d materials, and electrochemical energy storage communities.

Overall, I find the manuscript well written and the presentation of the results well suited for publication in Nature Communications. I was nonetheless confused about a few things I encourage the authors to address to make the manuscript more accessible and comprehensible in particular for non-experts in scanning transmission electron microscopy.

We are grateful to the reviewer for the positive comment.

1) How did the authors determine the bilayer nature of the graphene samples studied based on their data? Specifically, how did the authors rule out the samples studied were trilayer graphene where AM single layers could sit in neighboring interlayer spaces?

Response: The BLG was synthesized on the surface of either Cu (ACS Nano 7, 2587-2594 (2013)) or Ni-Cu alloy (Chem. Mater. 28, 4583-4592 (2016)) by using CVD method. The number of synthesized graphene layers was determined by the reflecting color contrast after transferring to a SiO₂/Si substrate, along with corresponding Raman spectroscopy. The quality of the BLG sample was extensively studied, and growth parameters were fine-tuned to achieve nearly 95% BLG coverage with less than 1% of trilayer or multilayer graphene in the grown samples. In TEM observations, the number of graphene layers can be precisely determined based on the ADF contrast and the diffraction spots, especially when two graphene layers stack with a twist angle, as shown in **Supplementary Figure 1**.

Supplementary Fig. 1 | STEM characterization of BLG and intercalated Cs layer. (a) ADF image of Cs intercalation in BLG. (b) FFT image from the BLG region in which the twist angle of BLG is 3°. (c) FFT image from the region of C₆Cs₂C₆. (d) ADF profile from the white box in (a). The ADF intensity of BLG is double that of SLG.

2) How did the authors determine the AM bilayers studied were confined in the interlayer gap between two layers of graphene, rather than located on top of or below these based on their data?

Response: This aspect is indeed what we have long been attempting to elucidate in this study. Essentially, we determine whether the AM bilayer resides in the interlayer gap or adheres to the graphene surface based on structural stability under e-beam irradiation and spectroscopy. The AMs adsorbed on the surface are mostly oxidized and amorphous; even if they initially form a crystalline structure, the surface AM layers are simply wiped out by the e-beam. However, AMs in BLG show lattice defects such as multiple vacancies during prolonged scanning e-beam observation, as demonstrated in

Supplementary Figure 9. The intercalated AM bilayers are consistently confined within the region of the BLG domain after the e-beam damage to the sample (see also **Supplementary Figure 1**), suggesting that the AM bilayer is confined in the interlayer gap between two graphene layers. The related discussion has been added to the revised manuscript.

Supplementary Fig. 9 | E-beam irradiation induced damages to the intercalated Cs bilayer. (a-d) Sequential ADF image of Cs sustained under the scanning e-beam observation for more than 2 minutes, with damaged Cs confined within the BLG domain, suggesting as an intercalated structure.

3) I am not an expert in ADF imaging, and was confused by the presentation of the data in Fig. 1. I expected to first of all see imaging/diffraction evidence for bilayer graphene without Cs, only then to be introduced to C6Cs2C6 in a second step. Instead, Figure 1 is all about C6Cs2C6 and the reader is left to deduce that for some reason the carbon atoms are not visible, instead the contrast mainly stems from Cs, the diffraction pattern itself would be different for bilayer graphene without Cs, etc. I recommend the authors to introduce what is known first, which could include merging Extended Data Fig. 3 with Fig. 1 for example.

Response: We appreciate the valuable comment from the reviewer. As suggested, we agree that presenting both the BLG and Cs-intercalated regions in the same figure will make it easier for readers to understand the structure. In response to this feedback, we have up updated Figure 1 by merging it with Extended Data Fig. 3, as presented below.

Figure 1 | STEM-ADF image and atomic structure of Cs bilayer in BLG ($C_6Cs_2C_6$). **a**, A STEM-ADF image of Cs-intercalated BLG displaying the $C_6Cs_2C_6$ structure. **b,c**, FFT image of BLG and Cs domains in **(a)**. BLG exhibits a twist angle of 2.7° between its two graphene layers. The [100] and [110] spots of $C_6Cs_2C_6$ domain correspond to interplanar d-spacings of 3.70 \AA and 2.14 \AA in real space, with the Cs [110] spots overlapping with the graphene [100] spots. **d**, Top-view atomic model of $C_6Cs_2C_6$. The yellow rhombus highlights a graphene (1×1) unit cell with $a = b = 2.46 \text{ \AA}$, while the red rhombus highlights the unit cell of the Cs lattice with $(\sqrt{3} \times \sqrt{3})R30^\circ$ lattice, where $a = b = 4.26 \text{ \AA}$. **e**, Side and perspective view of the $C_6Cs_2C_6$. The Cs atoms in different atomic planes are color-coded with two different shades of red. **f**, ADF profile of the hexagonal Cs layer along the pink and light green boxes in **(a)**. The red profile displays the shortest distance between two Cs atoms as 0.25 nm , consistent with the atomic model shown in **(d)**. The green profile shows the shorter distance between the Cs atoms, indicating the lateral displacement of the two Cs atomic plans.

4) Regarding Extended Data Fig. 3: Did the authors determine the 2.7 degree twist angle between graphene layers from the diffraction pattern? If so, is there also a 2.7 degree twist in the $C_6Cs_2C_6$ region? Doesn't the $(\sqrt{3} \times \sqrt{3}) R 30^\circ$ structure necessitate AA stacking of the graphene layers or does that supercell refer to only one of the two graphene layers?

Response: We appreciate the insightful question raised by the reviewer regarding the influence of the twist angle in BLG on the stacking structure of $C_6Cs_2C_6$. Indeed, we have observed that the domain size of the Cs bilayer is dependent on the twist angle of BLG. Specifically, in cases where BLG exhibits AA or AB stacking, the Cs bilayer can uniformly form hcp stacking. As the twist angle of BLG decreases ($<10^\circ$), the hcp Cs bilayer domain size becomes smaller, leading to the appearance of lattice-displaced domains. Conversely, in cases with larger BLG twist angles ($>10^\circ$), the hcp stacking Cs domains become even smaller and form periodic superlattice structures. Simply say, $(\sqrt{3} \times \sqrt{3}) R 30^\circ$ remains but the domain size is different according to the twist angle of BLG. The AM atoms find a balanced position between the potential of the outer graphene lattice and the metallic bonds between Cs atoms for hcp stacking.

To address this BLG twist angle-dependent phenomenon, we are actively developing corresponding theoretical models. Given the depth of this discussion, we have decided to present it as another full study in a separate paper, accompanied by a thorough discussion. The relevant comments have been added to the revised manuscript.

5) Could the authors expand their discussion by commenting on what they think is special about the 4 nm thickness beyond which the authors assert no more bilayer AMs would occur in graphite? Couldn't bilayer AMs still occur near the surface of an even thicker graphite flake, but maybe not near its center?

Response: The bilayer AM intercalation requires a significant expansion of the interlayer spacing within graphene layers, making it more likely to occur near the surface of graphite. We hypothesize that bilayer AMs predominantly occur near the surface of thick graphite, while single-layer AMs are more prominent in regions away from the surface, as the reviewer mentioned. The absence of visualizing bilayer AMs in thick graphite may be attributed to the fact that these bilayer AMs near the surface layers are out of focus in our STEM imaging when viewed from the top. We have included this discussion in the revised manuscript.

6) Could the authors expand their discussion by commenting on why they think only bilayers stable? What do the authors think is the mechanism that rules out the formation of trilayers or thicker AM layers in their experiment once the two graphene sheets have been separated?

Response: We believe that the metallic bond and the charge transfer phenomenon of the AM are the key factors in determining the stability of formation and the number of intercalated layers in a vdW gap. The low electronegativity nature of AM tends to donate electron when in contact with other elements, contributing to stabilization. In the intercalation of two AM layers, both layers can undergo charge transfer to the outer graphene layers. However, if trilayers or more layers of AM are intercalated, no charge transfer will occur in the middle layers of AM, result in less energetic favorability compared to the bilayer case.

The theoretical reference paper (Nano Energy, 75, 104927 (2020)) also suggests that the trilayer form of K, Rb, and Cs is less energetically favorable than the bilayer structure when intercalated in BLG. Additionally, an increasing number of AM layers also results in less favorability in formation energy in BLG. A related discussion has been added to the revised manuscript.

Minor comments:

- 1) Line 192: “charge carrier plasmon from the bilayer AMs to BLG” may have to read “charge carrier transport from the bilayer AMs to BLG” instead.
- 2) Legends in Fig. 2 g-I all read “C6K2C6” which needs correction.
- 3) Illegibly small labels/axis numbers in Figs. 2 d-f (insets),

Response: We appreciate the reviewer’s corrections to enhance the clarity of the paper. All the comments and corresponding changes have been addressed in the revised manuscript.

Figure 2 | Structure and spectroscopy of K, Rb, and Cs bilayers in BLG. **a-c**, ADF image of $C_6K_2C_6$, $C_6Rb_2C_6$, and $C_6Cs_2C_6$ in BLG. **d**, The EELS core loss spectra for the K L-edge of $C_6K_2C_6$, amorphous oxidized compounds of K adsorbed on the surface of BLG, and K_2CO_3 . **e**, The EELS core loss spectra for the Rb L-edge of $C_6Rb_2C_6$, amorphous oxidized compounds of Rb adsorbed on the surface of BLG, and Rb_2CO_3 . **f**, The EELS core loss spectra for the Cs M-edge of $C_6Cs_2C_6$, amorphous oxidized compounds of Cs adsorbed on the surface of BLG, and Cs_2CO_3 . The oxygen K-edges for these materials are presented in the insets. **g-i**, EELS valence loss spectra of K, Rb, and Cs intercalated samples. The black lines represent the original valence loss spectra, while the colored lines depict the background-subtracted edges. The grey dashed lines serve as a reference spectrum of AB stacking BLG. The charge carrier plasmon in all three samples appears as a broad peak around 1.5~3 eV, as highlighted by the corresponding colors.

REVIEWERS' COMMENTS

Reviewer #1 (Remarks to the Author):

Most of my comments have been addressed properly. I agree with the publication.

Reviewer #2 (Remarks to the Author):

The authors have answered all my comments in a satisfactory way and I am pleased to recommend this manuscript for publication.

Reviewer #3 (Remarks to the Author):

I thank the authors for their detailed response and the changes made to the manuscript. The authors have addressed all my concerns. I recommend publication in Nature Communications.